# CHARACTERIZING SIGNAL PROPAGATION TO CLOSE THE PERFORMANCE GAP IN UNNORMALIZED RESNETS

**Andrew Brock, Soham De & Samuel L. Smith**
Deepmind
{ajbrock, sohamde, slsmith}@google.com

## ABSTRACT

Batch Normalization is a key component in almost all state-of-the-art image classifiers, but it also introduces practical challenges: it breaks the independence between training examples within a batch, can incur compute and memory overhead, and often results in unexpected bugs. Building on recent theoretical analyses of deep ResNets at initialization, we propose a simple set of analysis tools to characterize signal propagation on the forward pass, and leverage these tools to design highly performant ResNets without activation normalization layers. Crucial to our success is an adapted version of the recently proposed Weight Standardization. Our analysis tools show how this technique preserves the signal in networks with ReLU or Swish activation functions by ensuring that the per-channel activation means do not grow with depth. Across a range of FLOP budgets, our networks attain performance competitive with the state-of-the-art EfficientNets on ImageNet. Our code is available at `http://dpmd.ai/nfnets`.

## 1 INTRODUCTION

BatchNorm has become a core computational primitive in deep learning (Ioffe & Szegedy, 2015), and it is used in almost all state-of-the-art image classifiers (Tan & Le, 2019; Wei et al., 2020). A number of different benefits of BatchNorm have been identified. It smoothens the loss landscape (Santurkar et al., 2018), which allows training with larger learning rates (Bjorck et al., 2018), and the noise arising from the minibatch estimates of the batch statistics introduces implicit regularization (Luo et al., 2019). Crucially, recent theoretical work (Balduzzi et al., 2017; De & Smith, 2020) has demonstrated that BatchNorm ensures good signal propagation at initialization in deep residual networks with identity skip connections (He et al., 2016b;a), and this benefit has enabled practitioners to train deep ResNets with hundreds or even thousands of layers (Zhang et al., 2019).

However, BatchNorm also has many disadvantages. Its behavior is strongly dependent on the batch size, performing poorly when the per device batch size is too small or too large (Hoffer et al., 2017), and it introduces a discrepancy between the behaviour of the model during training and at inference time. BatchNorm also adds memory overhead (Rota Bulò et al., 2018), and is a common source of implementation errors (Pham et al., 2019). In addition, it is often difficult to replicate batch normalized models trained on different hardware. A number of alternative normalization layers have been proposed (Ba et al., 2016; Wu & He, 2018), but typically these alternatives generalize poorly or introduce their own drawbacks, such as added compute costs at inference.

Another line of work has sought to eliminate layers which normalize hidden activations entirely. A common trend is to initialize residual branches to output zeros (Goyal et al., 2017; Zhang et al., 2019; De & Smith, 2020; Bachlechner et al., 2020), which ensures that the signal is dominated by the skip path early in training. However while this strategy enables us to train deep ResNets with thousands of layers, it still degrades generalization when compared to well-tuned baselines (De & Smith, 2020). These simple initialization strategies are also not applicable to more complicated architectures like EfficientNets (Tan & Le, 2019), the current state of the art on ImageNet (Russakovsky et al., 2015).

This work seeks to establish a general recipe for training deep ResNets without normalization layers, which achieve test accuracy competitive with the state of the art. Our contributions are as follows:

- We introduce Signal Propagation Plots (SPPs): a simple set of visualizations which help us inspect signal propagation at initialization on the forward pass in deep residual networks. Leveraging these SPPs, we show how to design unnormalized ResNets which are constrained to have signal propagation properties similar to batch-normalized ResNets.

- We identify a key failure mode in unnormalized ResNets with ReLU or Swish activations and Gaussian weights. Because the mean output of these non-linearities is positive, the squared mean of the hidden activations on each channel grows rapidly as the network depth increases. To resolve this, we propose Scaled Weight Standardization, a minor modification of the recently proposed Weight Standardization (Qiao et al., 2019; Huang et al., 2017b), which prevents the growth in the mean signal, leading to a substantial boost in performance.

- We apply our normalization-free network structure in conjunction with Scaled Weight Standardization to ResNets on ImageNet, where we for the first time attain performance which is comparable or better than batch-normalized ResNets on networks as deep as 288 layers.

- Finally, we apply our normalization-free approach to the RegNet architecture (Radosavovic et al., 2020). By combining this architecture with the compound scaling strategy proposed by Tan & Le (2019), we develop a class of models without normalization layers which are competitive with the current ImageNet state of the art across a range of FLOP budgets.

## 2 BACKGROUND

**Deep ResNets at initialization:** The combination of BatchNorm (Ioffe & Szegedy, 2015) and skip connections (Srivastava et al., 2015; He et al., 2016a) has allowed practitioners to train deep ResNets with hundreds or thousands of layers. To understand this effect, a number of papers have analyzed signal propagation in normalized ResNets at initialization (Balduzzi et al., 2017; Yang et al., 2019). In a recent work, De & Smith (2020) showed that in normalized ResNets with Gaussian initialization, the activations on the $\ell^{th}$ residual branch are suppressed by factor of $O(\sqrt{\ell})$, relative to the scale of the activations on the skip path. This biases the residual blocks in deep ResNets towards the identity function at initialization, ensuring well-behaved gradients. In unnormalized networks, one can preserve this benefit by introducing a learnable scalar at the end of each residual branch, initialized to zero (Zhang et al., 2019; De & Smith, 2020; Bachlechner et al., 2020). This simple change is sufficient to train deep ResNets with thousands of layers without normalization. However, while this method is easy to implement and achieves excellent convergence on the training set, it still achieves lower test accuracies than normalized networks when compared to well-tuned baselines.

These insights from studies of batch-normalized ResNets are also supported by theoretical analyses of unnormalized networks (Taki, 2017; Yang & Schoenholz, 2017; Hanin & Rolnick, 2018; Qi et al., 2020). These works suggest that, in ResNets with identity skip connections, if the signal does not explode on the forward pass, the gradients will neither explode nor vanish on the backward pass. Hanin & Rolnick (2018) conclude that multiplying the hidden activations on the residual branch by a factor of $O(1/d)$ or less, where $d$ denotes the network depth, is sufficient to ensure trainability at initialization.

**Alternate normalizers:** To counteract the limitations of BatchNorm in different situations, a range of alternative normalization schemes have been proposed, each operating on different components of the hidden activations. These include LayerNorm (Ba et al., 2016), InstanceNorm (Ulyanov et al., 2016), GroupNorm (Wu & He, 2018), and many more (Huang et al., 2020). While these alternatives remove the dependency on the batch size and typically work better than BatchNorm for very small batch sizes, they also introduce limitations of their own, such as introducing additional computational costs during inference time. Furthermore for image classification, these alternatives still tend to achieve lower test accuracies than well-tuned BatchNorm baselines. As one exception, we note that the combination of GroupNorm with Weight Standardization (Qiao et al., 2019) was recently identified as a promising alternative to BatchNorm in ResNet-50 (Kolesnikov et al., 2019).

## 3 SIGNAL PROPAGATION PLOTS

Although papers have recently theoretically analyzed signal propagation in ResNets (see Section 2), practitioners rarely empirically evaluate the scales of the hidden activations at different depths in-

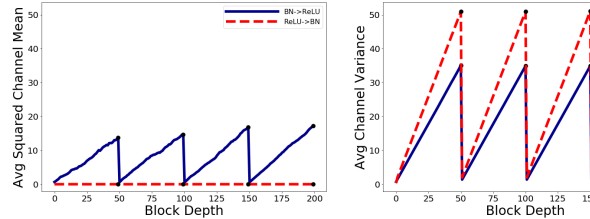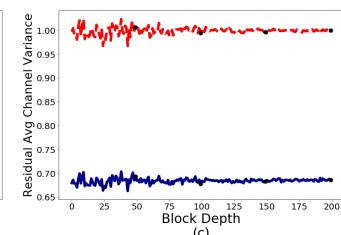

Figure 1: Signal Propagation Plot for a ResNetV2-600 at initialization with BatchNorm, ReLU activations and He init, in response to an $\mathcal{N}(0,1)$ input at 512px resolution. Black dots indicate the end of a stage. Blue plots use the BN-ReLU-Conv ordering while red plots use ReLU-BN-Conv.

side a specific deep network when designing new models or proposing modifications to existing architectures. By contrast, we have found that plotting the statistics of the hidden activations at different points inside a network, when conditioned on a batch of either random Gaussian inputs or real training examples, can be extremely beneficial. This practice both enables us to immediately detect hidden bugs in our implementation before launching an expensive training run destined to fail, and also allows us to identify surprising phenomena which might be challenging to derive from scratch.

We therefore propose to formalize this good practice by introducing Signal Propagation Plots (SPPs), a simple graphical method for visualizing signal propagation on the forward pass in deep ResNets. We assume identity residual blocks of the form $x_{\ell+1} = f_\ell(x_\ell) + x_\ell$, where $x_\ell$ denotes the input to the $\ell^{th}$ block and $f_\ell$ denotes the function computed by the $\ell^{th}$ residual branch. We consider 4-dimensional input and output tensors with dimensions denoted by $NHWC$, where $N$ denotes the batch dimension, $C$ denotes the channels, and $H$ and $W$ denote the two spatial dimensions. To generate SPPs, we initialize a single set of weights according to the network initialization scheme, and then provide the network with a batch of input examples sampled from a unit Gaussian distribution. Then, we plot the following hidden activation statistics at the output of each residual block:

- Average Channel Squared Mean, computed as the square of the mean across the $NHW$ axes, and then averaged across the $C$ axis. In a network with good signal propagation, we would expect the mean activations on each channel, averaged across a batch of examples, to be close to zero. Importantly, we note that it is necessary to measure the averaged squared value of the mean, since the means of different channels may have opposite signs.

- Average Channel Variance, computed by taking the channel variance across the $NHW$ axes, and then averaging across the $C$ axis. We generally find this to be the most informative measure of the signal magnitude, and to clearly show signal explosion or attenuation.

- Average Channel Variance on the end of the residual branch, before merging with the skip path. This helps assess whether the layers on the residual branch are correctly initialized.

We explore several other possible choices of statistics one could measure in Appendix G, but we have found these three to be the most informative. We also experiment with feeding the network real data samples instead of random noise, but find that this step does not meaningfully affect the key trends. We emphasize that SPPs do not capture every property of signal propagation, and they only consider the statistics of the forward pass. Despite this simplicity, SPPs are surprisingly useful for analyzing deep ResNets in practice. We speculate that this may be because in ResNets, as discussed in Section 2 (Taki, 2017; Yang & Schoenholz, 2017; Hanin & Rolnick, 2018), the backward pass will typically neither explode nor vanish so long as the signal on the forward pass is well behaved.

As an example, in Figure 1 we present the SPP for a 600-layer pre-activation ResNet (He et al., 2016a)[1] with BatchNorm, ReLU activations, and He initialization (He et al., 2015). We compare the standard BN-ReLU-Conv ordering to the less common ReLU-BN-Conv ordering. Immediately, several key patterns emerge. First, we note that the Average Channel Variance grows linearly with the depth in a given stage, and resets at each transition block to a fixed value close to 1. The linear growth arises because, at initialization, the variance of the activations satisfy $\text{Var}(x_{\ell+1}) = \text{Var}(x_\ell) + \text{Var}(f_\ell(x_\ell))$, while BatchNorm ensures that the variance of the activations at the end

---

[1]See Appendix E for an overview of ResNet blocks and their order of operations.

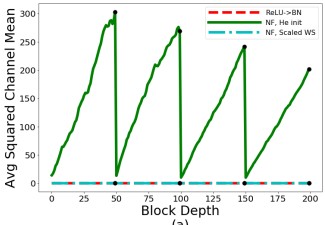 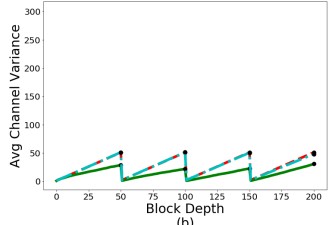 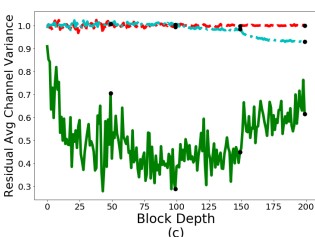

Figure 2: SPPs for three different variants of the ResNetV2-600 network (with ReLU activations). In red, we show a batch normalized network with ReLU-BN-Conv ordering. In green we show a normalizer-free network with He-init and $\alpha = 1$. In cyan, we show the same normalizer-free network but with Scaled Weight Standardization. We note that the SPPs for a normalizer-free network with Scaled Weight Standardization are almost identical to those for the batch normalized network.

of each residual branch is independent of depth (De & Smith, 2020). The variance is reset at each transition block because in these blocks the skip connection is replaced by a convolution operating on a normalized input, undoing any signal growth on the skip path in the preceding blocks.

With the BN-ReLU-Conv ordering, the Average Squared Channel Means display similar behavior, growing linearly with depth between transition blocks. This may seem surprising, since we expect BatchNorm to center the activations. However with this ordering the final convolution on a residual branch receives a rectified input with positive mean. As we show in the following section, this causes the outputs of the branch on any single channel to also have non-zero mean, and explains why $\mathrm{Var}(f_\ell(x_\ell)) \approx 0.68$ for all depths $\ell$. Although this "mean-shift" is explicitly counteracted by the normalization layers in subsequent residual branches, it will have serious consequences when attempting to remove normalization layers, as discussed below. In contrast, the ReLU-BN-Conv ordering trains equally stably while avoiding this mean-shift issue, with $\mathrm{Var}(f_\ell(x_\ell)) \approx 1$ for all $\ell$.

# 4 NORMALIZER-FREE RESNETS (NF-RESNETS)

With SPPs in hand to aid our analysis, we now seek to develop ResNet variants without normalization layers, which have good signal propagation, are stable during training, and reach test accuracies competitive with batch-normalized ResNets. We begin with two observations from Section 3. First, for standard initializations, BatchNorm downscales the input to each residual block by a factor proportional to the standard deviation of the input signal (De & Smith, 2020). Second, each residual block increases the variance of the signal by an approximately constant factor. We propose to mimic this effect by using residual blocks of the form $x_{\ell+1} = x_\ell + \alpha f_\ell(x_\ell/\beta_\ell)$, where $x_\ell$ denotes the input to the $\ell^{th}$ residual block and $f_\ell(\cdot)$ denotes the $\ell^{th}$ residual branch. We design the network such that:

- $f(\cdot)$, the function computed by the residual branch, is parameterized to be variance preserving at initialization, i.e., $\mathrm{Var}(f_\ell(z)) = \mathrm{Var}(z)$ for all $\ell$. This constraint enables us to reason about the signal growth in the network, and estimate the variances analytically.

- $\beta_\ell$ is a fixed scalar, chosen as $\sqrt{\mathrm{Var}(x_\ell)}$, the expected empirical standard deviation of the activations $x_\ell$ at initialization. This ensures the input to $f_\ell(\cdot)$ has unit variance.

- $\alpha$ is a scalar hyperparameter which controls the rate of variance growth between blocks.

We compute the expected empirical variance at residual block $\ell$ analytically according to $\mathrm{Var}(x_\ell) = \mathrm{Var}(x_{\ell-1}) + \alpha^2$, with an initial expected variance of $\mathrm{Var}(x_0) = 1$, and we set $\beta_\ell = \sqrt{\mathrm{Var}(x_\ell)}$. A similar approach was proposed by Arpit et al. (2016) for non-residual networks. As noted in Section 3, the signal variance in normalized ResNets is reset at each transition layer due to the shortcut convolution receiving a normalized input. We mimic this reset by having the shortcut convolution in transition layers operate on $(x_\ell/\beta_\ell)$ rather than $x_\ell$, ensuring unit signal variance at the start of each stage ($\mathrm{Var}(x_{\ell+1}) = 1 + \alpha^2$ following each transition layer). For simplicity, we call residual networks employing this simple scaling strategy *Normalizer-Free ResNets* (NF-ResNets).

### 4.1 ReLU Activations Induce Mean Shifts

We plot the SPPs for Normalizer-Free ResNets (NF-ResNets) with $\alpha = 1$ in Figure 2. In green, we consider a NF-ResNet, which initializes the convolutions with Gaussian weights using He initialization (He et al., 2015). Although one might expect this simple recipe to be sufficient to achieve good signal propagation, we observe two unexpected features in practice. First, the average value of the squared channel mean grows rapidly with depth, achieving large values which exceed the average channel variance. This indicates a large "mean shift", whereby the hidden activations for different training inputs (in this case different vectors sampled from the unit normal) are strongly correlated (Jacot et al., 2019; Ruff et al., 2019). Second, as observed for BN-ReLU-Conv networks in Section 3, the scale of the empirical variances on the residual branch are consistently smaller than one.

To identify the origin of these effects, in Figure 7 (in Appendix F) we provide a similar SPP for a linearized version of ResNetV2-600 without ReLU activation functions. When the ReLU activations are removed, the averaged squared channel means remain close to zero for all block depths, and the empirical variance on the residual branch fluctuates around one. This motivates the following question: why might ReLU activations cause the scale of the mean activations on a channel to grow?

To develop an intuition for this phenomenon, consider the transformation $z = Wg(x)$, where $W$ is arbitrary and fixed, and $g(\cdot)$ is an activation function that acts component-wise on iid inputs $x$ such that $g(x)$ is also iid. Thus, $g(\cdot)$ can be any popular activation function like ReLU, tanh, SiLU, etc. Let $\mathbb{E}(g(x_i)) = \mu_g$ and $\mathrm{Var}(g(x_i)) = \sigma_g^2$ for all dimensions $i$. It is straightforward to show that the expected value and the variance of any single unit $i$ of the output $z_i = \sum_j^N W_{i,j} g(x_j)$ is given by:

$$\mathbb{E}(z_i) = N\mu_g\mu_{W_{i,\cdot}}, \quad \text{and} \quad \mathrm{Var}(z_i) = N\sigma_g^2(\sigma_{W_{i,\cdot}}^2 + \mu_{W_{i,\cdot}}^2), \tag{1}$$

where $\mu_{W_{i,\cdot}}$ and $\sigma_{W_{i,\cdot}}$ are the mean and standard deviation of the $i^{th}$ row of $W$:

$$\mu_{W_{i,\cdot}} = \frac{1}{N}\sum_j^N W_{i,j}, \quad \text{and} \quad \sigma_{W_{i,\cdot}}^2 = \frac{1}{N}\sum_j^N W_{i,j}^2 - \mu_{W_{i,\cdot}}^2. \tag{2}$$

Now consider $g(\cdot)$ to be the ReLU activation function, i.e., $g(x) = \max(x, 0)$. Then $g(x) \geq 0$, which implies that the input to the linear layer has positive mean (ignoring the edge case when all inputs are less than or equal to zero). In particular, notice that if $x_i \sim \mathcal{N}(0, 1)$ for all $i$, then $\mu_g = 1/\sqrt{2\pi}$. Since we know that $\mu_g > 0$, if $\mu_{W_{i,\cdot}}$ is also non-zero, then the output of the transformation, $z_i$, will also exhibit a non-zero mean. Crucially, even if we sample $W$ from a distribution centred around zero, any specific weight matrix drawn from this distribution will almost surely have a non-zero empirical mean, and consequently the outputs of the residual branches on any specific channel will have non-zero mean values. This simple NF-ResNet model with He-initialized weights is therefore often unstable, and it is increasingly difficult to train as the depth increases.

### 4.2 Scaled Weight Standardization

To prevent the emergence of a mean shift, and to ensure that the residual branch $f_\ell(\cdot)$ is variance preserving, we propose Scaled Weight Standardization, a minor modification of the recently proposed Weight Standardization (Qiao et al., 2019) which is also closely related to Centered Weight Normalization (Huang et al., 2017b). We re-parameterize the convolutional layers, by imposing,

$$\hat{W}_{i,j} = \gamma \cdot \frac{W_{i,j} - \mu_{W_{i,\cdot}}}{\sigma_{W_{i,\cdot}}\sqrt{N}}, \tag{3}$$

where the mean $\mu$ and variance $\sigma$ are computed across the fan-in extent of the convolutional filters. We initialize the underlying parameters $W$ from Gaussian weights, while $\gamma$ is a fixed constant. As in Qiao et al. (2019), we impose this constraint throughout training as a differentiable operation in the forward pass of the network. Recalling equation 1, we can immediately see that the output of the transformation using Scaled WS, $z = \hat{W}g(x)$, has expected value $\mathbb{E}(z_i) = 0$ for all $i$, thus eliminating the mean shift. Furthermore, the variance $\mathrm{Var}(z_i) = \gamma^2\sigma_g^2$, meaning that for a correctly chosen $\gamma$, which depends on the non-linearity $g$, the layer will be variance preserving. Scaled Weight Standardization is cheap during training and free at inference, scales well (with the number of parameters rather than activations), introduces no dependence between batch elements and no discrepancy in training and test behavior, and its implementation does not differ in distributed training. These desirable properties make it a compelling alternative for replacing BatchNorm.

The SPP of a normalizer-free ResNet-600 employing Scaled WS is shown in Figure 2 in cyan. As we can see, Scaled Weight Standardization eliminates the growth of the average channel squared mean at initialization. Indeed, the SPPs are almost identical to the SPPs for a batch-normalized network employing the ReLU-BN-Conv ordering, shown in red. Note that we select the constant $\gamma$ to ensure that the channel variance on the residual branch is close to one (discussed further below). The variance on the residual branch decays slightly near the end of the network due to zero padding.

### 4.3 DETERMINING NONLINEARITY-SPECIFIC CONSTANTS $\gamma$

The final ingredient we need is to determine the value of the gain $\gamma$, in order to ensure that the variances of the hidden activations on the residual branch are close to 1 at initialization. Note that the value of $\gamma$ will depend on the specific nonlinearity used in the network. We derive the value of $\gamma$ by assuming that the input $x$ to the nonlinearity is sampled iid from $\mathcal{N}(0,1)$. For ReLU networks, this implies that the outputs $g(x) = \max(x,0)$ will be sampled from the rectified Gaussian distribution with variance $\sigma_g^2 = (1/2)(1 - (1/\pi))$ (Arpit et al., 2016). Since $\text{Var}(\hat{W}g(x)) = \gamma^2\sigma_g^2$, we set $\gamma = 1/\sigma_g = \frac{\sqrt{2}}{\sqrt{1-\frac{1}{\pi}}}$ to ensure that $\text{Var}(\hat{W}g(x)) = 1$. While the assumption $x \sim \mathcal{N}(0,1)$ is not typically true unless the network width is large, we find this approximation to work well in practice.

For simple nonlinearities like ReLU or tanh, the analytical variance of the non-linearity $g(x)$ when $x$ is drawn from the unit normal may be known or easy to derive. For other nonlinearities, such as SiLU ((Elfwing et al., 2018; Hendrycks & Gimpel, 2016), recently popularized as Swish (Ramachandran et al., 2018)), analytically determining the variance can involve solving difficult integrals, or may even not have an analytical form. In practice, we find that it is sufficient to numerically approximate this value by the simple procedure of drawing many $N$ dimensional vectors $x$ from the Gaussian distribution, computing the empirical variance $\text{Var}(g(x))$ for each vector, and taking the square root of the average of this empirical variance. We provide an example in Appendix D showing how this can be accomplished for any nonlinearity with just a few lines of code and provide reference values.

### 4.4 OTHER BUILDING BLOCKS AND RELAXED CONSTRAINTS

Our method generally requires that any additional operations used in a network maintain good signal propagation, which means many common building blocks must be modified. As with selecting $\gamma$ values, the necessary modification can be determined analytically or empirically. For example, the popular Squeeze-and-Excitation operation (S+E, Hu et al. (2018)), $y = sigmoid(MLP(pool(h))) * h$, involves multiplication by an activation in $[0,1]$, and will tend to attenuate the signal and make the model unstable. This attenuation can clearly be seen in the SPP of a normalizer-free ResNet using these blocks (see Figure 9 in Appendix F). If we examine this operation in isolation using our simple numerical procedure explained above, we find that the expected variance is 0.5 (for unit normal inputs), indicating that we simply need to multiply the output by 2 to recover good signal propagation. We empirically verified that this simple change is sufficient to restore training stability.

In practice, we find that either a similarly simple modification to any given operation is sufficient to maintain good signal propagation, or that the network is sufficiently robust to the degradation induced by the operation to train well without modification. We also explore the degree to which we can relax our constraints and still maintain stable training. As an example of this, to recover some of the expressivity of a normal convolution, we introduce learnable affine gains and biases to the Scaled WS layer (the gain is applied to the weight, while the bias is added to the activation, as is typical). While we could constrain these values to enforce good signal propagation by, for example, downscaling the output by a scalar proportional to the values of the gains, we find that this is not necessary for stable training, and that stability is not impacted when these parameters vary freely. Relatedly, we find that using a learnable scalar multiplier at the end of the residual branch initialized to 0 (Goyal et al., 2017; De & Smith, 2020) helps when training networks over 150 layers, even if we ignore this modification when computing $\beta_\ell$. In our final models, we employ several such relaxations without loss of training stability. We provide detailed explanations for each operation and any modifications we make in Appendix C (also detailed in our model code in Appendix D).

### 4.5 SUMMARY

In summary, the core recipe for a Normalizer-Free ResNet (NF-ResNet) is:

1. Compute and forward propagate the expected signal variance $\beta_\ell^2$, which grows by $\alpha^2$ after each residual block ($\beta_0 = 1$). Downscale the input to each residual branch by $\beta_\ell$.

2. Additionally, downscale the input to the convolution on the skip path in transition blocks by $\beta_\ell$, and reset $\beta_{\ell+1} = 1 + \alpha^2$ following a transition block.

3. Employ Scaled Weight Standardization in all convolutional layers, computing $\gamma$, the gain specific to the activation function $g(x)$, as the reciprocal of the expected standard deviation, $\frac{1}{\sqrt{\text{Var}(g(x))}}$, assuming $x \sim \mathcal{N}(0, 1)$.

Code is provided in Appendix D for a reference Normalizer-Free Network.

## 5 EXPERIMENTS

### 5.1 AN EMPIRICAL EVALUATION ON RESNETS

|  | FixUp | | SkipInit | | NF-ResNets (ours) | | BN-ResNets | |
|---|---|---|---|---|---|---|---|---|
|  | Unreg. | Reg. | Unreg. | Reg. | Unreg. | Reg. | Unreg. | Reg. |
| RN50 | $74.0 \pm .5$ | $75.9 \pm .3$ | $73.7 \pm .2$ | $75.8 \pm .2$ | $75.8 \pm .1$ | $\mathbf{76.8 \pm .1}$ | $\mathbf{76.8 \pm .1}$ | $76.4 \pm .1$ |
| RN101 | $75.4 \pm .6$ | $77.6 \pm .3$ | $75.1 \pm .1$ | $77.3 \pm .2$ | $77.1 \pm .1$ | $\mathbf{78.4 \pm .1}$ | $78.0 \pm .1$ | $78.1 \pm .1$ |
| RN152 | $75.8 \pm .4$ | $78.4 \pm .3$ | $75.7 \pm .2$ | $78.0 \pm .1$ | $77.6 \pm .1$ | $\mathbf{79.1 \pm .1}$ | $78.6 \pm .2$ | $78.8 \pm .1$ |
| RN200 | $76.2 \pm .5$ | $78.7 \pm .3$ | $75.9 \pm .2$ | $78.2 \pm .1$ | $77.9 \pm .2$ | $\mathbf{79.6 \pm .1}$ | $79.0 \pm .2$ | $79.2 \pm .1$ |
| RN288 | $76.2 \pm .4$ | $78.4 \pm .4$ | $76.3 \pm .2$ | $78.7 \pm .2$ | $78.1 \pm .1^*$ | $\mathbf{79.5 \pm .1}$ | $78.8 \pm .1$ | $\mathbf{79.5 \pm .1}$ |

Table 1: ImageNet Top-1 Accuracy (%) for ResNets with FixUp (Zhang et al., 2019) or SkipInit (De & Smith, 2020), Normalizer-Free ResNets (ours), and Batch-Normalized ResNets. We train all variants both with and without additional regularization (stochastic depth and dropout). Results are given as the median accuracy $\pm$ the standard deviation across 5 random seeds. $^*$ indicates a setting where two runs collapsed and results are reported only using the 3 seeds which train successfully.

We begin by investigating the performance of Normalizer-Free pre-activation ResNets on the ILSVRC dataset (Russakovsky et al., 2015), for which we compare our networks to FixUp initialization (Zhang et al., 2019), SkipInit (De & Smith, 2020), and batch-normalized ResNets. We use a training setup based on Goyal et al. (2017), and train our models using SGD (Robbins & Monro, 1951) with Nesterov's Momentum (Nesterov, 1983; Sutskever et al., 2013) for 90 epochs with a batch size of 1024 and a learning rate which warms up from zero to 0.4 over the first 5 epochs, then decays to zero using cosine annealing (Loshchilov & Hutter, 2017). We employ standard baseline preprocessing (sampling and resizing distorted bounding boxes, along with random flips), weight decay of 5e-5, and label smoothing of 0.1 (Szegedy et al., 2016). For Normalizer-Free ResNets (NF-ResNets), we chose $\alpha = 0.2$ based on a small sweep, and employ SkipInit as discussed above. For both FixUp and SkipInit we had to reduce the learning rate to 0.2 to enable stable training.

We find that without additional regularization, our NF-ResNets achieve higher training accuracies but lower test accuracies than their batch-normalized counterparts. This is likely caused by the known regularization effect of BatchNorm (Hoffer et al., 2017; Luo et al., 2019; De & Smith, 2020). We therefore introduce stochastic depth (Huang et al., 2016) with a rate of 0.1, and Dropout (Srivastava et al., 2014) before the final linear layer with a drop probability of 0.25. We note that adding this same regularization does not substantially improve the performance of the normalized ResNets in our setup, suggesting that BatchNorm is indeed already providing some regularization benefit.

In Table 1 we compare performance of our networks (NF-ResNets) against the baseline (BN-ResNets), across a wide range of network depths. After introducing additional regularization, NF-ResNets achieve performance better than FixUp/SkipInit and competitive with BN across all network depths, with our regularized NF-ResNet-288 achieving top-1 accuracy of 79.5%. However, some of the 288 layer normalizer-free models undergo training collapse at the chosen learning rate,

| | NF-ResNets (ours) | | | BN-ResNets | | |
|---|---|---|---|---|---|---|
| | BS=1024 | BS=8 | BS=4 | BS=1024 | BS=8 | BS=4 |
| ResNet-50 | $69.9 \pm 0.1$ | $\mathbf{69.6 \pm 0.1}$ | $\mathbf{69.9 \pm 0.1}$ | $\mathbf{70.9 \pm 0.1}$ | $65.7 \pm 0.2$ | $55.7 \pm 0.3$ |

Table 2: ImageNet Top-1 Accuracy (%) for Normalizer-Free ResNets and Batch-Normalized ResNet-50s on ImageNet, using very small batch sizes trained for 15 epochs. Results are given as the median accuracy $\pm$ the standard deviation across 5 random seeds. Performance degrades severely for Batch-Normalized networks, while Normalizer-Free ResNets retain good performance.

but only when unregularized. While we can remove this instability by reducing the learning rate to 0.2, this comes at the cost of test accuracy. We investigate this failure mode in Appendix A.

One important limitation of BatchNorm is that its performance degrades when the per-device batch size is small (Hoffer et al., 2017; Wu & He, 2018). To demonstrate that our normalizer-free models overcome this limitation, we train ResNet-50s on ImageNet using very small batch sizes of 8 and 4, and report the results in Table 2. These models are trained for 15 epochs (4.8M and 2.4M training steps, respectively) with a learning rate of 0.025 for batch size 8 and 0.01 for batch size 4. For comparison, we also include the accuracy obtained when training for 15 epochs at batch size 1024 and learning rate 0.4. The NF-ResNet achieves significantly better performance when the batch size is small, and is not affected by the shift from batch size 8 to 4, demonstrating the usefulness of our approach in the microbatch setting. Note that we do not apply stochastic depth or dropout in these experiments, which may explain superior performance of the BN-ResNet at batch size 1024. We also study the transferability of our learned representations to the downstream tasks of semantic segmentation and depth estimation, and present the results of these experiments in Appendix H.

## 5.2 DESIGNING PERFORMANT NORMALIZER-FREE NETWORKS

We now turn our attention to developing unnormalized networks which are competitive with the state-of-the-art EfficientNet model family across a range of FLOP budgets (Tan & Le, 2019), We focus primarily on the small budget regime (EfficientNets B0-B4), but also report results for B5 and hope to extend our investigation to larger variants in future work.

First, we apply Scaled WS and our Normalizer-Free structure directly to the EfficientNet backbone.[2] While we succeed in training these networks stably without normalization, we find that even after extensive tuning our Normalizer-Free EfficientNets still substantially underperform their batch-normalized baselines. For example, our normalization free B0 variant achieves 73.5% top-1, a 3.2% absolute degradation relative to the baseline. We hypothesize that this degradation arises because Weight Standardization imposes a very strong constraint on depth-wise convolutions (which have an input channel count of 1), and this constraint may remove a substantial fraction of the model expressivity. To support this claim, we note that removing Scaled WS from the depth-wise convolutions improves the test accuracy of Normalizer-Free EfficientNets, although this also reduces the training stability.

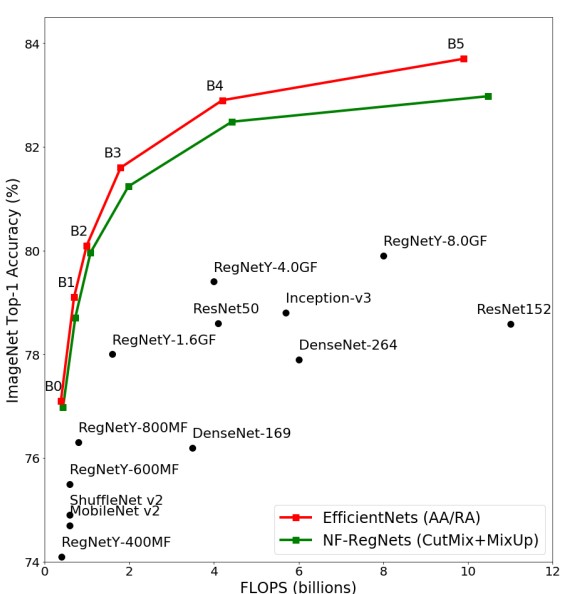

Figure 3: ImageNet Top-1 test accuracy versus FLOPs.

[2]We were unable to train EfficientNets using SkipInit (De & Smith, 2020; Bachlechner et al., 2020). We speculate this may be because the EfficientNet backbone contains both residual and non-residual components.

Therefore, to overcome the potentially
poor interactions between Weight Stan-
dardization and depth-wise convolutions,
we decided to instead study Normalizer-Free variants of the RegNet model family (Radosavovic
et al., 2020). RegNets are slightly modified variants of ResNeXts (Xie et al., 2017), developed via
manual architecture search. Crucially, RegNets employ grouped convolutions, which we anticipate
are more compatible with Scaled WS than depth-wise convolutions, since the fraction of the degrees
of freedom in the model weights remaining after the weight standardization operation is higher.

We develop a new base model by taking the 0.4B FLOP RegNet variant, and making several mi-
nor architectural changes which cumulatively substantially improve the model performance. We
describe our final model in full in Appendix C, however we emphasize that most of the architec-
ture changes we introduce simply reflect well-known best practices from the literature (Tan & Le,
2019; He et al., 2019). To assess the performance of our Normalizer-Free RegNets across a range of
FLOPS budgets, we apply the EfficientNet compound scaling approach (which increases the width,
depth and input resolution in tandem according to a set of three power laws learned using architecture
search) to obtain model variants at a range of approximate FLOPS targets. Denoting these models
NF-RegNets, we train variants B0-B5 (analogous to EfficientNet variants) using both baseline pre-
processing and combined CutMix (Yun et al., 2019) and MixUp (Zhang et al., 2018) augmentation.
Note that we follow the same compound scaling hyper-parameters used by EfficientNets, and do not
retune these hyper-parameters on our own architecture. We compare the test accuracies of Efficient-
Nets and NF-RegNets on ImageNet in Figure 3, and we provide the corresponding numerical values
in Table 3 of Appendix A. We present a comparison of training speeds in Table 5 of Appendix A.

For each FLOPS and augmentation setting, NF-RegNets attain comparable but slightly lower test
accuracies than EfficientNets, while being substantially faster to train. In the augmented setting, we
report EfficientNet results with AutoAugment (AA) or RandAugment (RA), (Cubuk et al., 2019;
2020), which we find performs better than training EfficientNets with CutMix+MixUp. However,
both AA and RA degrade the performance and stability of NF-RegNets, and hence we report results
of NF-RegNets with CutMix+Mixup instead. We hypothesize that this occurs because AA and
RA were developed by applying architecture search on batch-normalized models, and that they may
therefore change the statistics of the dataset in a way that negatively impacts signal propagation when
normalization layers are removed. To support this claim, we note that inserting a single BatchNorm
layer after the first convolution in an NF-RegNet removes these instabilities and enables us to train
stably with either AA or RA, although this approach does not achieve higher test set accuracies.

These observations highlight that, although our models do benefit from most of the architectural
improvements and best practices which researchers have developed from the hundreds of thousands
of device hours used while tuning batch-normalized models, there are certain aspects of existing
state-of-the-art models, like AA and RA, which may implicitly rely on the presence of activation
normalization layers in the network. Furthermore there may be other components, like depth-wise
convolutions, which are incompatible with promising new primitives like Weight Standardization.
It is therefore inevitable that some fine-tuning and model development is necessary to achieve com-
petitive accuracies when removing a component like batch normalization which is crucial to the
performance of existing state-of-the-art networks. Our experiments confirm for the first time that
it is possible to develop deep ResNets which do not require batch normalization or other activation
normalization layers, and which not only train stably and achieve low training losses, but also attain
test accuracy competitive with the current state of the art on a challenging benchmark like ImageNet.

# 6 CONCLUSION

We introduce Normalizer-Free Networks, a simple approach for designing residual networks which
do not require activation normalization layers. Across a range of FLOP budgets, our models achieve
performance competitive with the state-of-the-art EfficientNets on ImageNet. Meanwhile, our em-
pirical analysis of signal propagation suggests that batch normalization resolves two key failure
modes at initialization in deep ResNets. First, it suppresses the scale of the hidden activations on
the residual branch, preventing signal explosion. Second, it prevents the mean squared scale of the
activations on each channel from exceeding the variance of the activations between examples. Our
Normalizer-Free Networks were carefully designed to resolve both of these failure modes.

ACKNOWLEDGMENTS

We would like to thank Karen Simonyan for helpful discussions and guidance, as well as Guillaume Desjardins, Michael Figurnov, Nikolay Savinov, Omar Rivasplata, Relja Arandjelović, and Rishub Jain.

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

# APPENDIX A    EXPERIMENT DETAILS

| Model | #FLOPs | #Params | Top-1 w/o Augs | Top-1 w/ Augs |
|---|---|---|---|---|
| NF-RegNet-B0 | 0.44B | 8.3M | $76.8 \pm 0.2$ | $77.0 \pm 0.1$ |
| EfficientNet-B0 | 0.39B | 5.3M | 76.7 | 77.1 |
| RegNetY-400MF | 0.40B | 4.3M | 74.1 | − |
| NF-RegNet-B1 | 0.73B | 9.8M | $78.6 \pm 0.1$ | $78.7 \pm 0.1$ |
| EfficientNet-B1 | 0.70B | 7.8M | 78.7 | 79.1 |
| RegNetY-600MF | 0.60B | 6.1M | 75.5 | − |
| RegNetY-800MF | 0.80B | 6.3M | 76.3 | − |
| MobileNet (Howard et al., 2017) | 0.60B | 4.2M | 70.6 | − |
| MobileNet v2 (Sandler et al., 2018) | 0.59B | 6.9M | 74.7 | − |
| ShuffleNet v2 (Ma et al., 2018) | 0.59B | − | 74.9 | − |
| NF-RegNet-B2 | 1.09B | 13.4M | $79.6 \pm 0.1$ | $80.0 \pm 0.1$ |
| EfficientNet-B2 | 1.00B | 9.2M | 79.8 | 80.1 |
| NF-RegNet-B3 | 1.98B | 17.6M | $80.6 \pm 0.1$ | $81.2 \pm 0.1$ |
| EfficientNet-B3 | 1.80B | 12.0M | 81.1 | 81.6 |
| NF-RegNet-B4 | 4.43B | 28.5M | $81.7 \pm 0.1$ | $82.5 \pm 0.1$ |
| EfficientNet-B4 | 4.20B | 19.0M | 82.5 | 82.9 |
| RegNetY-4.0GF | 4.00B | 20.6M | 79.4 | − |
| ResNet50 | 4.10B | 26.0M | 76.8 | 78.6 |
| DenseNet-169 (Huang et al., 2017a) | 3.50B | 14.0M | 76.2 | − |
| NF-RegNet-B5 | 10.48B | 47.5M | $82.0 \pm 0.2$ | $83.0 \pm 0.2$ |
| EfficientNet-B5 | 9.90B | 30.0M | 83.1 | 83.7 |
| RegNetY-12GF | 12.10B | 51.8M | 80.3 | − |
| ResNet152 | 11.00B | 60.0M | 78.6 | − |
| Inception-v4 (Szegedy et al., 2017) | 13.00B | 48.0M | 80.0 | − |

Table 3:  ImageNet Top-1 Accuracy (%) comparison for NF-RegNets and recent state-of-the-art models. "w/ Augs" refers to accuracy with advanced augmentations: for EfficientNets, this is with AutoAugment or RandAugment.  For NF-RegNets, this is with CutMix + MixUp.  NF-RegNet results are reported as the median and standard deviation across 5 random seeds.

## A.1    STABILITY, LEARNING RATES, AND BATCH SIZES

Previous work (Goyal et al., 2017) has established a fairly robust linear relationship between the optimal learning rate (or highest stable learning rate) and batch size for Batch-Normalized ResNets. As noted in Smith et al. (2020), we also find that this relationship breaks down past batch size 1024 for our unnormalized ResNets, as opposed to 2048 or 4096 for normalized ResNets. Both the optimal learning rate and the highest stable learning rate decrease for higher batch sizes. This also appears to correlate with depth: when not regularized, our deepest models are not always stable with a learning rate of 0.4. While we can mitigate this collapse by reducing the learning rate for deeper nets, this introduces additional tuning expense and is clearly undesirable. It is not presently clear why regularization aids in stability; we leave investigation of this phenomenon to future work.

Taking a closer look at collapsed networks, we find that even though their outputs have exploded (becoming large enough to go NaN), their weight magnitudes are not especially large, even if we remove our relaxed affine transforms and train networks whose layers are purely weight-standardized. The singular values of the weights, however, end up poorly conditioned, meaning that the Lipschitz constant of the network can become quite large, an effect which Scaled WS does not prevent. One might consider adopting one of the many techniques from the GAN literature to regularize or constrain this constant (Gulrajani et al., 2017; Miyato et al., 2018), but we have found that this added complexity and expense is not necessary to develop performant unnormalized networks.

This collapse highlights an important limitation of our approach, and of SPPs: as SPPs only show signal prop for a given state of a network (i.e., at initialization), no guarantees are provided far from initialization. This fact drives us to prefer parameterizations like Scaled WS rather than solely relying on initialization strategies, and highlights that while good signal propagation is generally necessary for stable optimization, it is not always sufficient.

## A.2 Training Speed

| | BS16 | | BS32 | | BS64 | |
|---|---|---|---|---|---|---|
| | NF | BN | NF | BN | NF | BN |
| ResNet-50 | 17.3 | 16.42 | 10.5 | 9.45 | 5.79 | 5.24 |
| ResNet-101 | 10.4 | 9.4 | 6.28 | 5.75 | 3.46 | 3.08 |
| ResNet-152 | 7.02 | 5.57 | 4.4 | 3.95 | 2.4 | 2.17 |
| ResNet-200 | 5.22 | 3.53 | 3.26 | 2.61 | OOM | OOM |
| ResNet-288 | 3.0 | 2.25 | OOM | OOM | OOM | OOM |

Table 4: Training speed (in training iterations per second) comparisons of NF-ResNets and BN-ResNets on a single 16GB V100 for various batch sizes.

| | BS16 | | BS32 | | BS64 | |
|---|---|---|---|---|---|---|
| | NF-RegNet | EffNet | NF-RegNet | EffNet | NF-RegNet | EffNet |
| B0 | 12.2 | 5.23 | 9.25 | 3.61 | 6.51 | 2.61 |
| B1 | 9.06 | 3.0 | 6.5 | 2.14 | 4.69 | 1.55 |
| B2 | 5.84 | 2.22 | 4.05 | 1.68 | 2.7 | 1.16 |
| B3 | 4.49 | 1.57 | 3.1 | 1.05 | 2.13 | OOM |
| B4 | 2.73 | 0.94 | 1.96 | OOM | OOM | OOM |
| B5 | 1.66 | OOM | OOM | OOM | OOM | OOM |

Table 5: Training speed (in training iterations per second) comparisons of NF-RegNets and Batch-Normalized EfficientNets on a single 16GB V100 for various batch sizes.

We evaluate the relative training speed of our normalizer-free models against batch-normalized models by comparing training speed (measured as the number of training steps per second). For comparing NF-RegNets against EfficientNets, we measure using the EfficientNet image sizes for each variant to employ comparable settings, but in practice we employ smaller image sizes so that our actual observed training speed for NF-RegNets is faster.

## APPENDIX B    MODIFIED BUILDING BLOCKS

In order to maintain good signal propagation in our Normalizer-Free models, we must ensure that any architectural modifications do not compromise our model's conditioning, as we cannot rely on activation normalizers to automatically correct for such changes. However, our models are not so fragile as to be unable to handle slight relaxations in this realm. We leverage this robustness to improve model expressivity and to incorporate known best practices for model design.

### B.1    AFFINE GAINS AND BIASES

First, we add affine gains and biases to our network, similar to those used by activation normalizers. These are applied as a vector gain, each element of which multiplies a given output unit of a reparameterized convolutional weight, and a vector bias, which is added to the output of each convolution. We also experimented with using these as a separate affine transform applied before the ReLU, but moved the parameters next to the weight instead to enable constant-folding for inference. As is common practice with normalizer parameters, we do not weight decay or otherwise regularize these weights.

Even though these parameters are allowed to vary freely, we do not find that they are responsible for training instability, even in networks where we observe collapse. Indeed, we find that for settings which collapse (typically due to learning rates being too high), removing the affine transform has no impact on stability. As discussed in Appendix A, we observe that model instability arises as a result of the collapse of the spectra of the weights, rather than any consequence of the affine gains and biases.

### B.2    STOCHASTIC DEPTH

We incorporate Stochastic Depth (Huang et al., 2016), where the output of the residual branch of a block is randomly set to zero during training. This is often implemented such that if the block is kept, its value is divided by the keep probability. We remove this rescaling factor to help maintain signal propagation when the signal is kept, but otherwise do not find it necessary to modify this block.

While it is possible that we might have an example where many blocks are dropped and signals are attenuated, in practice we find that, as with affine gains, removing Stochastic Depth does not improve stability, and adding it does not reduce stability. One might also consider a slightly more principled variant of Stochastic Depth in this context, where the skip connection is upscaled by $1+\alpha$ if the residual branch is dropped, resulting in the variance growing as expected, but we did not find this strategy necessary.

### B.3    SQUEEZE AND EXCITE LAYERS

As mentioned in Section 4.4, we incorporate Squeeze and Excitation layers (Hu et al., 2018), which we empirically find to reduce signal magnitude by a factor of 0.5, which is simply corrected by multiplying by 2. This was determined using a similar procedure to that used to find $\gamma$ values for a given nonlinearity, as demonstrated in Appendix D. We validate this empirically by training NF-RegNet models with unmodified S+E blocks, which do not train stably, and NF-RegNet models with the additional correcting factor of 2, which do train stably.

### B.4    AVERAGE POOLING

In line with best practices determined by He et al. (2019), in our NF-RegNet models we replace the strided 1x1 convolutions with average pooling followed by 1x1 convolutions, a common alternative also employed in Zagoruyko & Komodakis (2016). We found that average pooling with a kernel of size $k \times k$ tended to attenuate the signal by a factor of $k$, but that it was not necessary to apply any correction due to this. While this will result in mis-estimation of $\beta$ values at initialization, it does not harm training (and average pooling in fact improved results over strided 1x1 convolutions in every case we tried), so we simply include this operation as-is.

## APPENDIX C    MODEL DETAILS

We develop the NF-RegNet architecture starting with a RegNetY-400MF architecture (Radosavovic et al. (2020)) a low-latency RegNet variant which also uses Squeeze+Excite blocks (Hu et al., 2018)) and uses grouped convolutions with a group width of 8. Following EfficientNets, we first add an additional expansion convolution after the final residual block, expanding to $1280w$ channels, where $w$ is a model width multiplier hyperparameter. We find this to be very important for performance: if the classifier layer does not have access to a large enough feature basis, it will tend to underfit (as measured by higher training losses) and underperform. We also experimented with adding an additional linear expansion layer after the global average pooling, but found this not to provide the same benefit.

Next, we replace the strided 1x1 convolutions in transition layers with average pooling followed by 1x1 convolutions (following He et al. (2019)), which we also find to improve performance. We switch from ReLU activations to SiLU activations (Elfwing et al., 2018; Hendrycks & Gimpel, 2016; Ramachandran et al., 2018). We find that SiLU's benefits are only realized when used in conjunction with EMA (the model averaging we use, explained below), as in EfficientNets. The performance of the underlying weights does not seem to be affected by the difference in nonlinearities, so the improvement appears to come from SiLU apparently being more amenable to averaging.

We then tune the choice of width $w$ and bottleneck ratio $g$ by sweeping them on the 0.4B FLOP model. Contrary to Radosavovic et al. (2020) which found that inverted bottlenecks (Sandler et al., 2018) were not performant, we find that inverted bottlenecks strongly outperformed their compressive bottleneck counterparts, and select $w = 0.75$ and $g = 2.25$. Following EfficientNets (Tan & Le, 2019), the very first residual block in a network uses $g = 1$, a FLOP-reducing strategy that does not appear to harmfully impact performance.

We also modify the S+E layers to be wider by making their hidden channel width a function of the block's expanded width, rather than the block's input width (which is smaller in an inverted bottleneck). This results in our models having higher parameter counts than their equivalent FLOP target EfficientNets, but has minimal effect on FLOPS, while improving performance. While both FLOPS and parameter count play a part in the latency of a deployed model, (the quantity which is often most relevant in practice) neither are fully predictive of latency (Xiong et al., 2020). We choose to focus on the FLOPS target instead of parameter count, as one can typically obtain large improvements in accuracy at a given parameter count by, for example, increasing the resolution of the input image, which will dramatically increase the FLOPS.

With our baseline model in hand, we apply the EfficientNet compound scaling (increasing width, depth, and input image resolution) to obtain a family of models at approximately the same FLOP targets as each EfficientNet variant. We directly use the EfficientNet width and depth multipliers for models B0 through B5, and tune the test image resolution to attain similar FLOP counts (although our models tend to have slightly higher FLOP budgets). Again contrary to Radosavovic et al. (2020), which scales models almost entirely by increasing width and group width, we find that the Efficient-Net compound scaling works effectively as originally reported, particularly with respect to image size. Improvements might be made by applying further architecture search, such as tuning the $w$ and $g$ values for each variant separately, or by choosing the group width separately for each variant.

Following Touvron et al. (2019), we train on images of slightly lower resolution than we test on, primarily to reduce the resource costs of training. We do not employ the fine-tuning procedure of Touvron et al. (2019). The exact train and test image sizes we use are visible in our model code in Appendix D.

We train using SGD with Nesterov Momentum, using a batch size of 1024 for 360 epochs, which is chosen to be in line with EfficientNet's schedule of 360 epoch training at batch size 4096. We employ a 5 epoch warmup to a learning rate of 0.4 (Goyal et al., 2017), and cosine annealing to 0 over the remaining epochs (Loshchilov & Hutter, 2017). As with EfficientNets, we also take an exponential moving average of the weights (Polyak, 1964), using a decay of 0.99999 which employs a warmup schedule such that at iteration $i$, the decay is $decay = min(i, \frac{1+i}{10+i})$. We choose a larger decay than the EfficientNets value of 0.9999, as EfficientNets also take an EMA of the running average statistics of the BatchNorm layers, resulting in a longer horizon for the averaged model.

As with EfficientNets, we find that some of our models attain their best performance before the end of training, but unlike EfficientNets we do not employ early stopping, instead simply reporting performance at the end of training. The source of this phenomenon is that as some models (particularly larger models) reach the end of their decay schedule, the rate of change of their weights slows, ultimately resulting in the averaged weights converging back towards the underlying (less performant) weights. Future work in this area might consider examining the interaction between averaging and learning rate schedules.

Following EfficientNets, we also use stochastic depth (modified to remove the rescaling by the keep rate, so as to better preserve signal) with a drop rate that scales from 0 to 0.1 with depth (reduced from the EfficientNets value of 0.2). We swept this value and found the model to not be especially sensitive to it as long as it was not chosen beyond 0.25. We apply Dropout (Srivastava et al., 2014) to the final pooled activations, using the same Dropout rates as EfficientNets for each variant. We also use label smoothing (Szegedy et al., 2016) of 0.1, and weight decay of 5e-5.

## APPENDIX D    MODEL CODE

We here provide reference code using Numpy (Harris et al., 2020) and JAX (Bradbury et al., 2018). Our full training code is publicly available at `dpmd.ai/nfnets`.

### D.1    NUMERICAL APPROXIMATIONS OF NONLINEARITY-SPECIFIC GAINS

It is often faster to determine the nonlinearity-specific constants $\gamma$ empirically, especially when the chosen activation functions are complex or difficult to integrate. One simple way to do this is for the SiLU function is to sample many (say, 1024) random C-dimensional vectors (of say size 256) and compute the average variance, which will allow for computing an estimate of the constant. Empirically estimating constants to ensure good signal propagation in networks at initialization has previously been proposed in Mishkin & Matas (2016) and Kingma & Dhariwal (2018).

```
import jax
import jax.numpy as jnp
key = jax.random.PRNGKey(2) # Arbitrary key
# Produce a large batch of random noise vectors
x = jax.random.normal(key, (1024, 256))
y = jax.nn.silu(x)
# Take the average variance of many random batches
gamma = jnp.mean(jnp.var(y, axis=1)) ** -0.5
```

## APPENDIX E   OVERVIEW OF EXISTING BLOCKS

This appendix contains an overview of several different types of residual blocks.

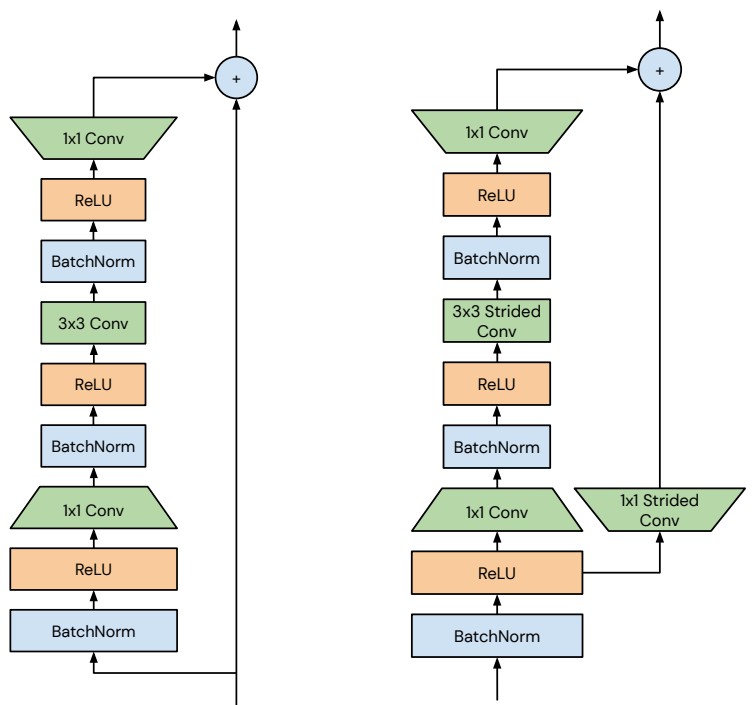

(a) Pre-Activation ResNet Block          (b) Pre-Activation ResNet Transition Block

Figure 4: Residual Blocks for pre-activation ResNets (He et al., 2016a). Note that some variants swap the order of the nonlinearity and the BatchNorm, resulting in signal propagation which is more similar to that of our normalizer-free networks.

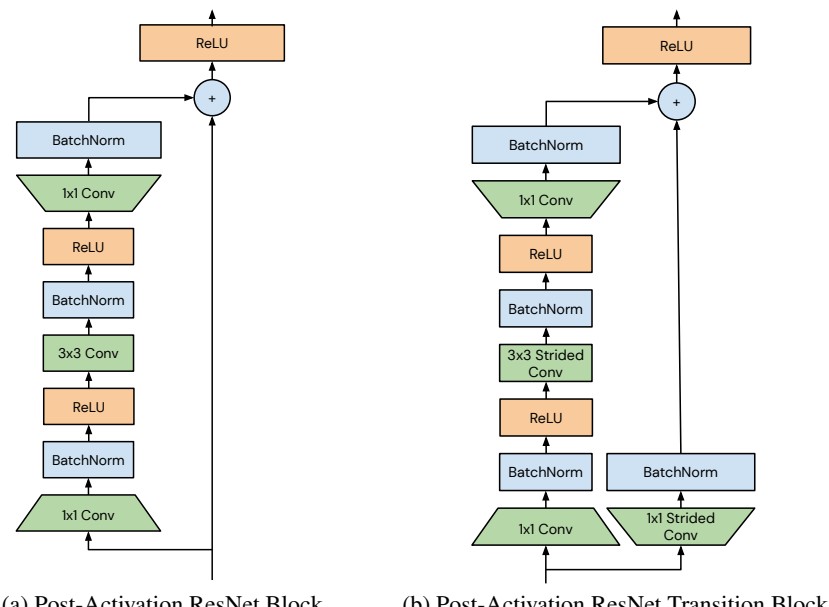

(a) Post-Activation ResNet Block      (b) Post-Activation ResNet Transition Block

Figure 5: Residual Blocks for post-activation (original) ResNets (He et al., 2016b).

## APPENDIX F  ADDITIONAL SPPS

In this appendix, we include additional Signal Propagation Plots. For reference, given an $NHWC$ tensor, we compute the measured properties using the equivalent of the following Numpy (Harris et al., 2020) snippets:

- Average Channel Mean Squared:
  ```
  np.mean(np.mean(y, axis=[0, 1, 2]) ** 2)
  ```
- Average Channel Variance:
  ```
  np.mean(np.var(y, axis=[0, 1, 2]))
  ```
- Residual Average Channel Variance:
  ```
  np.mean(np.var(f(x), axis=[0, 1, 2]))
  ```

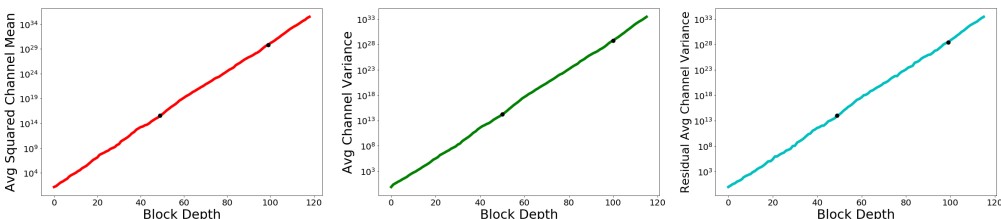

Figure 6: Signal Propagation Plot for a ResNetV2-600 with ReLU and He initialization, without any normalization, on a semilog scale. The scales of all three properties grow logarithmically due to signal explosion.

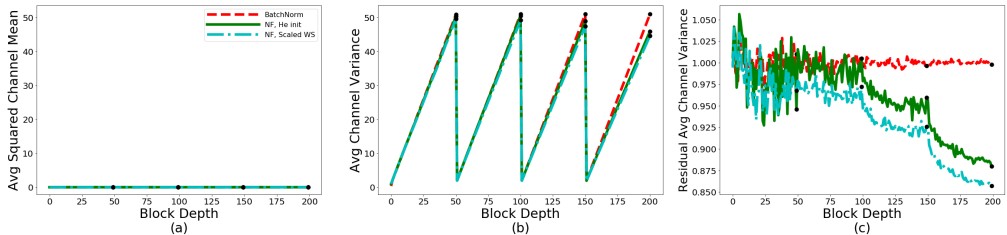

Figure 7: Signal Propagation Plot for 600-layer ResNetV2s with linear activations, comparing BatchNorm against with normalizer-free scaling. Note that the max-pooling operation in the ResNet stem has here been removed so that the inputs to the first blocks are centered.

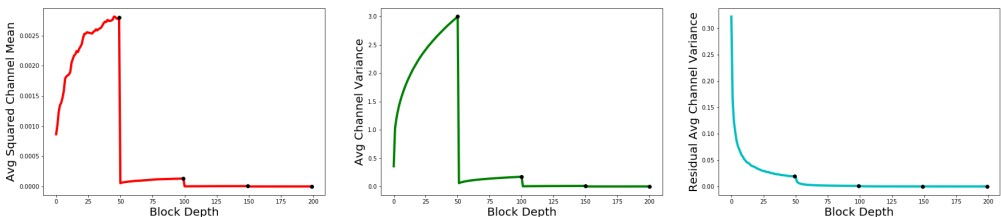

Figure 8: Signal Propagation Plot for a Normalizer-Free ResNetV2-600 with ReLU and Scaled WS, using $\gamma = \sqrt{2}$, the gain for ReLU from (He et al., 2015). As this gain (derived from $\sqrt{\frac{1}{\mathbb{E}[g(x)^2]}}$) is lower than the correct gain ($\sqrt{\frac{1}{Var(g(x))}}$), signals attenuate progressively in the first stage, then are further downscaled at each transition which uses a $\beta$ value that assumes a higher incoming scale.

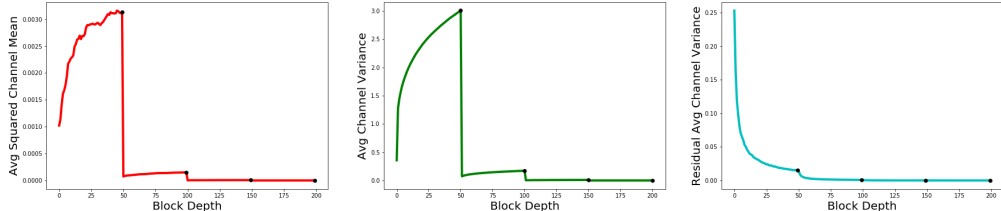

Figure 9: Signal Propagation Plot for a Normalizer-Free ResNetV2-600 with ReLU, Scaled WS with correctly chosen gains, and unmodified Squeeze-and-Excite Blocks. Similar to understimating $\gamma$ values, unmodified S+E blocks will attenuate the signal.

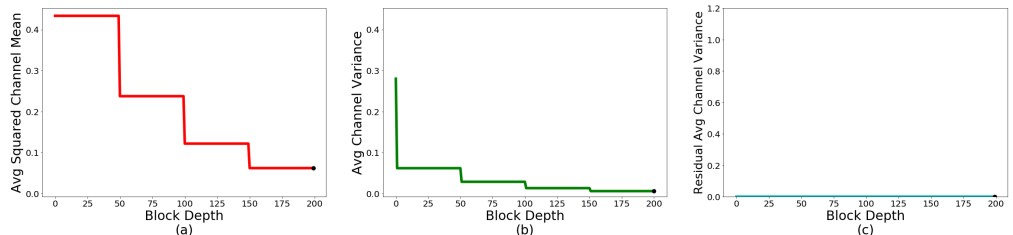

Figure 10: Signal Propagation Plot for a ResNet-600 with FixUp. Due to the zero-initialized weights, FixUp networks have constant variance in a stage, and still demonstrate variance resets across stages.

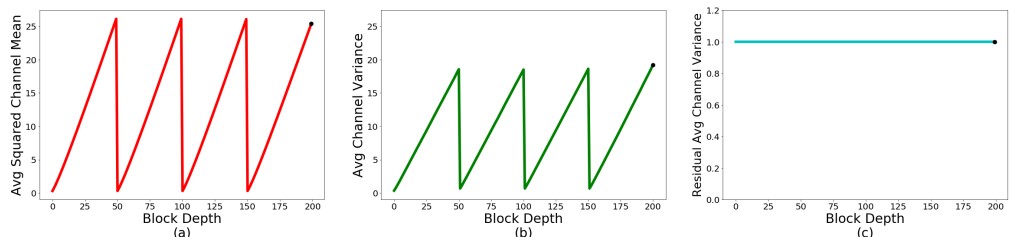

Figure 11: Signal Propagation Plot for a ResNet-600 V1 with post-activation ordering. As this variant applies BatchNorm at the end of the residual block, the Residual Average Channel Variance is kept constant at 1 throughout the model. This ordering also applies BatchNorm on shortcut 1x1 convolutions at stage transition, and thus also displays variance resets.

## APPENDIX G  NEGATIVE RESULTS

### G.1  FORWARD MODE VS DECOUPLED WS

Parameterization methods like Weight Standardization (Qiao et al., 2019), Weight Normalization (Salimans & Kingma, 2016), and Spectral Normalization (Miyato et al., 2018) are typically proposed as "foward mode" modifications applied to parameters during the forward pass of a network. This has two consequences: first, this means that the gradients with respect to the underlying parameters are influenced by the parameterization, and that the weights which are optimized may differ substantially from the weights which are actually plugged into the network.

One alternative approach is to implement "decoupled" variants of these parameterizers, by applying them as a projection step in the optimizer. For example, "Decoupled Weight Standardization" can be implemented atop any gradient based optimizer by replacing $W$ with the normalized $\hat{W}$ after the update step. Most papers proposing parameterizations (including the above) argue that the parameterization's gradient influence is helpful for learning, but this is typically argued with respect to simply ignoring the parameterization during the backward pass, rather than with respect to a strategy such as this.

Using a Forward-Mode parameterization may result in interesting interactions with moving averages or weight decay. For example, with WS, if one takes a moving average of the underlying weights, then applies the WS parameterization to the averaged weights, this will produce different results than if one took the EMA of the Weight-Standardized parameters. Weight decay will have a similar phenomenon: if one is weight decaying a parameter which is actually a proxy for a weight-standardized parameter, how does this change the behavior of the regularization?

We experimented with Decoupled WS and found that it reduced sensitivity to weight decay (presumably because of the strength of the projection step) and often improved the accuracy of the EMA weights early in training, but ultimately led to worse performance than using the originally proposed "forward-mode" formulation. We emphasize that our experiments in this regime were only cursory, and suggest that future work might seek to analyze these interactions in more depth.

We also tried applying Scaled WS as a regularizer ("Soft WS") by penalizing the mean squared error between the parameter $W$ and its Scaled WS parameterization, $\hat{W}$. We implemented this as a direct addition to the parameters following Loshchilov & Hutter (2019) rather than as a differentiated loss, with a scale hyperparameter controlling the strength of the regularization. We found that this scale could not be meaningfully decreased from its maximal value without drastic training instability, indicating that relaxing the WS constraint is better done through other means, such as the affine gains and biases we employ.

### G.2  MISCELLANEOUS

- For SPPs, we initially explored plotting activation mean (`np.mean(h)`) instead of the average squared channel mean, but found that this was less informative.

- We also initially explored plotting the average pixel norm: the Frobenius norm of each pixel (reduced across the $C$ axis) then averaged across the $NHW$ axis, `np.mean(np.linalg.norm(h, axis=-1)))`. We found that this value did not add any information not already contained in the channel or residual variance measures, and was harder to interpret due to it varying with the channel count.

- We explored NF-ResNet variants which maintained *constant* signal variance, rather than mimicking Batch-Normalized ResNets with signal growth + resets. The first of two key components in this approach was making use of "rescaled sum junctions," where the sum junction in a residual block was rewritten to downscale the shortcut path as $y = \frac{\alpha * f(x) + x}{\alpha^2}$, which is approximately norm-preserving if $f(x)$ is orthogonal to $x$ (which we observed to generally hold in practice). Instead of Scaled WS, this variant employed SeLU (Klambauer et al., 2017) activations, which we found to work as-advertised in encouraging centering and good scaling.

  While these networks could be made to train stably, we found tuning them to be difficult and were not able to easily recover the performance of BN-ResNets as we were with the approach ultimately presented in this paper.

## APPENDIX H    EXPERIMENTS WITH ADDITIONAL TASKS

### H.1    SEMANTIC SEGMENTATION ON PASCAL VOC

|  | NF-ResNets (ours) | BN-ResNets |
|---|---|---|
|  | mIoU | mIoU |
| ResNet-50 | 74.4 | 75.4 |
| ResNet-101 | 76.7 | 77.0 |
| ResNet-152 | 77.6 | 77.9 |
| ResNet-200 | 78.4 | 78.0 |

Table 6:  Results on Pascal VOC Semantic Segmentation.

We present additional results investigating the transferability of our normalizer-free models to downstream tasks, beginning with the Pascal VOC Semantic Segmentation task. We use the FCN architecture (Long et al., 2015) following He et al. (2020) and Grill et al. (2020) . We take the ResNet backbones of each variant and modify the 3x3 convolutions in the final stage to have dilation 2 and stride of 1, then add two extra 3x3 convolutions with dilation of 6, and a final 1x1 convolution for classification. We train for 30000 steps at batch size 16 using SGD with Nesterov Momentum of 0.9, a learning rate of 0.003 which is reduced by a factor of 10 at 70% and 90% of training, and weight decay of 5e-5. Training images are augmented with random scaling in the range [0.5, 2.0]), random horizontal flips, and random crops. Results in mean Intersection over Union (mIoU) are reported in Table 6 on the val2012 set using a single 513 pixel center crop. We do not add any additional regularization such as stochastic depth or dropout. NF-ResNets obtain comparable performance to their BN-ResNet counterparts across all variants.

### H.2    DEPTH ESTIMATION ON NYU DEPTH V2

|  | Higher better | | | Lower Better | |
|---|---|---|---|---|---|
|  | pct. $< 1.25$ | pct. $< 1.25^2$ | pct. $< 1.25^3$ | rms | rel |
| NF-ResNet-50 | 81.9 | 95.9 | 98.9 | 0.572 | 0.141 |
| BN-ResNet-50 | 81.7 | 95.7 | 98.8 | 0.579 | 0.141 |
| NF-ResNet-101 | 82.7 | 96.4 | 99.0 | 0.564 | 0.136 |
| BN-ResNet-101 | 83.4 | 96.2 | 98.9 | 0.563 | 0.132 |
| NF-ResNet-152 | 83.2 | 96.4 | 99.1 | 0.558 | 0.135 |
| BN-ResNet-152 | 81.6 | 96.0 | 98.9 | 0.579 | 0.140 |
| NF-ResNet-200 | 84.0 | 96.7 | 99.2 | 0.552 | 0.130 |
| BN-ResNet-200 | 84.6 | 96.6 | 99.1 | 0.548 | 0.125 |

Table 7:  Results on NYUv2 Depth Estimation.

We next present results for depth estimation on the NYU v2 dataset (Silberman et al., 2012) using the protocol from (Laina et al., 2016). We downsample the images and center crop them to [304, 228] pixels, then randomly flip and apply several color augmentations: *grayscale* with probability 30%, *brightness* with a maximum difference of 0.1255, *saturation* with a random factor picked from [0.5, 1.5], and *Hue* with adjustment factor picked in [-0.2, 0.2]. We take the features from the final residual stage and feed them into the up-projection blocks from (Silberman et al., 2012), then train with a reverse Huber loss (Laina et al., 2016; Owen, 2007). We train for 7500 steps at batch size 256 using SGD with Nesterov Momentum of 0.9, a learning rate of 0.16, and cosine annealing. We report results in Table7 using five metrics commonly used for this task: the percentage of pixels where the magnitude of the relative error (taken as the ratio of the predicted depth and the ground truth, where the denominator is whichever is smaller) is below a certain threshold, as well as rootmean-squared and relative error (rms and rel). As with semantic segmentation, we do not apply any additional regularization, and find that our normalizer-free ResNets attain comparable performance across all model variants.

