# OpenReview forum: "Characterizing signal propagation to close the performance gap in unnormalized ResNets"
_ICLR.cc/2021/Conference — ICLR 2021 Poster_

### Official Review · AnonReviewer2 · 2020-10-28
**ConvNets without normalization**

**Rating:** 7
**Confidence:** 3

**Review:**

The paper proposes a novel, and mathematically well motivated initialization scheme for deep ResNet like models.
This is used to train batchnorm free Resnets that compare to their baseline.
They also train RegNet like models that are fairly close to EfficientNets in terms of performance.

Comments:

You list problems with batchnorm:
1) 'it breaks independence between training examples'
   Why is this a problem? You don't include any experiments showing your method works well in the small batch regime where batchnorm is problematic.
2) 'it is suprisingly expensive to compute'
    Batchnorm was widely adopted as it made networks train much faster. And there is no computational overhead at test time. Does your method allow models to be trained more quickly, i.e. how does accuracy compare after $N$ minutes of training time?
3) 'it often results in unexpected bugs'
   I don't understand this point. Batchnorm has lead to many state-of-the-art results. In contrast you have only managed to get good results by using large amounts of prior knowledge that was obtained from experiments on the same validation set and using batchnorm.

Section 5.1 Why is the training collapse of the 288 layer models not reflected in Table 1?

Regarding Figure 3: Why compare your method **with** data augmentation and EfficientNets **without**? That seems very misleading to me. Getting close to proper EfficientNet performance without batchnorm is a decent achievement. Why include an inferior, incomparable baseline?

--update---
I am upgrading to 7: Good paper accept; as my concerns have been addressed by the additional experiments.
The proposed method is not yet a drop in replacement for BatchNorm in general, but it can be useful in specific circumstances, i.e. small batch-size training.

---

> ### Author Response · Authors · 2020-11-16
> **Response to R2**
>
> We thank the reviewer for the constructive feedback. We have included a top-level response that addresses some of the reviewer’s comments as well as describes new experimental results we have now added to an updated version of the paper. We address additional specific comments below.
>
> > Why is breaking independence between training examples a problem? Small batch results?
>
> We have added results with small batch training to our draft, as described in our top level comment.
>
> There are both theoretical and practical limitations of breaking the independence between training examples. Most theoretical models of optimization on neural networks require the assumption that all training examples are independent, and analyzing optimization behaviour without this assumption becomes significantly more challenging for non-trivial neural network models.
>
> From a practical standpoint, introducing dependencies between training examples leads to a number of additional design considerations (or additional communication) when implementing BatchNorm on distributed systems (e.g., does one use per-device BatchNorm or accept the cost of cross-replica BatchNorm?). This issue is also specifically noted in setups like contrastive learning (MoCo and SimCLR) where it requires specific care in order to prevent breakdown of the contrastive objective.
>
>
> > “Surprisingly expensive to compute”, but no experiments. Do NF-Nets train faster than BatchNorm?
>
> The primary cases where BatchNorm’s compute cost is “surprising” is due to the memory overhead it can incur or in distributed training. For example, for training semantic segmentation models with large image sizes this cost can be substantial, and has driven the development of techniques like In-Place Activated BatchNorm [1], where the authors note that addressing this issue with BatchNorm can reduce memory consumption by up to 50% for some models. In the case where one requires cross-replica BatchNorm (as is used in EfficientNet training) one also incurs inter-device communication costs which will vary widely with the hardware.
>
> We have added training speed numbers to appendix A.2 comparing NF-ResNets against BN-ResNets on a single GPU; our models are mildly faster in this simple test-case, which does not result in a big difference in accuracy after N minutes of training. Our NFNets models are substantially faster than EfficientNets, but this is more due to depthwise convs in EfficientNets being very slow in practice. Given this, we have revised the statement in the abstract to remove “surprisingly” and instead use the more factual statement that BatchNorm “can incur compute and memory overhead.”
>
>
> > Unexpected bugs? NF-Nets build on large amount of prior knowledge.
>
> We thank the reviewer for noting our lack of citation on bugs stemming from BatchNorm. In our top-level comment titled “The trouble with BatchNorm”, we list a number of known examples, and have added a citation to prior academic work specifically noting the prevalence of bugs related to BatchNorm.
>
> We agree that BatchNorm has been an integral part of state-of-the-art models in a number of tasks over the past few years, and we certainly build on a lot of prior knowledge from these highly tuned normalized models. However, BatchNorm’s importance on this front does not mean that it comes without disadvantages (as evidenced by the large body of work attempting to develop alternative normalizers or remove normalization), and we would argue that its prevalence is largely driven by (a) alternative normalizers typically incurring costs at inference or performing worse and (b) previous methods that remove normalization like FixUp or SkipInit resulting in degraded generalization performance. Our motivation is therefore to build on the strong prior work (largely derived using batch-normalized models) to develop models that are sufficiently robust and performant to be of interest to the community while not having the disadvantages of BatchNorm.
>
>
> > Training collapse of 288 nets.
>
> This was an accidental omission; we have updated Table 1 to indicate that the values reported for these runs are computed over the random seeds which do not collapse.
>
> > Comparing method with augs to EffNets without augs? Why include EffNets without augs as a baseline?
>
> We have updated the pareto front figure to only include EffNets with Augmentations. Please note that our results table (table 3) contains both standard preprocessing and training with augmentations for both EfficientNets and NFNets across all FLOP targets for completeness.
>
>
> [1] Samuel Rota Bulò, Lorenzo Porzi, Peter Kontschieder. In-Place Activated BatchNorm for Memory-Optimized Training of DNNs. CVPR 2018
>
> [2] ICLR anonymous submission: Deconstructing the Regularization of BatchNorm. Link: https://openreview.net/pdf?id=d-XzF81Wg1.

---

### Official Review · AnonReviewer3 · 2020-10-28

**Rating:** 7
**Confidence:** 4

**Review:**

Summary:

The authors propose a set of visualization tools named SPPs for better monitoring the key indicators, including Average Channel Squared Mean, Average Channel Variance, etc., of hidden activations. With SPPs in hand, they find two key failure modes at initialization in deep ResNets and then develop Normalizer-Free ResNets using Scaled Weight Standardization, achieving competitive performance on ResNet-288 and Efficient-Net.

Strength:
--The overall idea makes sense and the proposed method of removing batch normalization can reduce computational resources and speed up computing greatly.
--The visualization of the key indicators may be helpful for understanding how batch normalization works and how to remove batch normalization.
--The experiment results seem that the proposed method achieves good performance in large-scale ResNets.

Weakness:
-- There is a lack of accuracy comparison with other removing batch normalization works.
-- It will be better to prove the effectiveness of the proposed method in more tasks, not only the classification task.

Comments:
(1)	The batch size you used for training is 1024. As you know, decreasing batch size is good for online training on portable devices. Can the proposed method decrease the batch size to a small value like 2/4/8 after removing batch normalization? And if it is difficult, can you give an explanation on how batch size affects the performance after removing batch normalization with the proposed method.
(2)	You have visualized the key indicators of the proposed initialization method. Can you give the visualization of the same indicators of other initialization methods like Fixup initialization and make some comparison? I think it’s within your ability.
(3)	It’s better to add some accuracy comparisons with other removing batch normalization works.

------------------------------------------------------------------------------------------------------------------------------------------------------------------------------------
UPDATE: The author has addressed most of my concerns, but regarding the motivations and the benefits for the community, I still keep my score.

---

> ### Author Response · Authors · 2020-11-16
> **Response to R3**
>
> We thank the reviewer for their positive feedback. We have incorporated all of this feedback into our draft (the new experimental results in our paper are described in detail in our top-level comment on OpenReview) and here respond to specific points.
>
> > Comparison to other alternatives to normalization
>
> We have now included comparisons with Fixup initialization and SkipInit initialization on ResNets of various depths in table 1. In all cases, NF-ResNets outperform both Fixup and SkipInit.
>
> > Small batch experiments
>
> We have updated the draft to include additional experiments on ResNets for small batch sizes (8 and 4) in table 2, and find that NF-ResNets behave much better than BN-ResNets in this case. In particular, when going from batch size 8  to batch size 4, NF-ResNets do not lose any performance, but BN-ResNets suffer severe performance drops.
>
> > SPPs with Fixup initialization
>
> We have added SPPs for models with FixUp and for ResNet-V1 (post-activation) models in appendix F.
>
> > Additional downstream tasks
>
> We have now added experiments on transfer from BN-ResNets and NF-ResNets to the downstream tasks of Pascal VOC semantic segmentation and NYUv2 depth estimation, and find that NF-ResNets perform comparably to their batch-normalized counterparts on both tasks.

---

### Official Review · AnonReviewer1 · 2020-10-29

**Rating:** 5
**Confidence:** 5

**Review:**

This paper proposes the signal propagation plot (spp) which is a tool for analyzing residual networks and analyzes ResNet with/without BN. Based on the investigation, the authors first provide ResNet results without normalization with the proposed scaled weight standardization. Furthermore, the authors provide a bunch of models that are competitive to EfficientNets based on RegNetY-400MF,  which seem to be highly tuned in terms of architecture design.

Pros)

1. This paper is well written and easy to follow.

2. The proposed SPP seems to be a good tool for analyzing a model.


Cons)

1. The authors seem to have failed to provide any reasons for needing normalization free ResNet over the original BN-ResNet.

2. It is not clear that the trained model without NF with SWS can be used as a backbone that can be directly applied to downstream tasks (e.g., object detection).

3. I don't think it was necessary to show competitive results with EfficientNets using the other baseline - RegNet.  Especially, the proposed models (which are compared with EfficientNets) are highly-tuned trying to surpass EfficientNets' accuracy. This type of paper would be better to be focused on investigating the characteristics of a network.

Comments)
1. The major problem of this paper is none of the advantages of NF with SWS are highlighted over BN, so it is hard to find any reasons for replacing BN with NF-SWS. I mean non of the disadvantages of BN are addressed by the proposed method.

2. SPP could enlighten that an unusual ReLU-BN-Conv ordering would have some benefits but why should a network without normalization mimick the SPP trend of ResNet? It is unnatural that a network without normalization should follow the ResNet's behavior.

3. From the equation x_{l+1}=x_l+ a * f (x_l/b_l) at p.4, the proposed approach eventually normalizing the feature even if a or b_l is fixed. Moreover, gamma in eq. (3) additionally scaling up or down the weight which ultimately gives an effect on the computed feature.

4. How did the authors compute gamma in eq.(3) in a training phase? Why the provided code contains a learnable gamma?

5. Why "zero padding" at p.5 affects the variance decay in the rightmost graph in Figure 2?

6. Please clarify why RegNetY-400MF chose it as the baseline. It is not clear that the authors pick RegNet and tune it highly and compared with EfficientNets. If the authors decided to use RegNets, then it is natural to use NF-SWS-RegNets are compared with the original RegNets without any big modifications as shown in the model details in the Appendix.

7. Did the authors use trained ResNetV2-600 or randomly-initialized model?

8. How does SPP goes on with the post-activation network which uses the original bottleneck block consists of Conv-BN-ReLU?

9. How can the resolution downsampling block in a ResNet affect averaged channel mean and variance?

10. Why the ResNet experiments are done with weight decay of 5e-5? The common knowledge of training ResNet is with weighing decay of 1e-4, so one may unconvinced the result because of the tuning.

11. Comparing EfficientNets that are trained without CutMix and Mixup (but used randaug or autoaug) with the proposed models with cutmix and mixup seems to be not fair.

---

> ### Author Response · Authors · 2020-11-16
> **Response to R1, Part 1 of 2**
>
> We thank the reviewer for their valuable suggestions and detailed comments. We have included a top-level response that addresses some of the reviewer’s comments as well as describes additional experimental results we have now added to an updated version of the paper. Here we present additional responses for R1.
>
>
> > “None of the disadvantages of BN are addressed by the proposed method”
>
> The proposed method overcomes the following disadvantages:
>
> BatchNorm introduces a dependence between elements in a batch. Normalizer-Free Networks with Scaled weight Standardization do not have this property. Please see our top-level response titled “The trouble with BatchNorm” for a discussion on how this batch-dependence can be problematic in models.
>
> This batch-dependence aspect of BatchNorm also means that it is not always clear how to implement it in distributed training. When one may have a low per-device batch size, BatchNorm can behave poorly or require the additional communication overhead of synced cross-replica BatchNorm. In contrast, our proposed method is completely invariant to the number of devices (its implementation does not change) and does not require cross-device communication in the forward pass.
>
> As requested by the reviewers, we have now added ResNet experiments in the small batch setting to the paper, where networks with BatchNorm perform very poorly, while NF-ResNets perform well.
>
> BatchNorm has different behavior at training and test time. Norm-Free Nets and Scaled Weight Standardization behave the same at training and testing time. Scaled WS also does not require the maintenance of running averages.
>
> Other alternatives that remove the train-test discrepancy (GroupNorm or LayerNorm) incur compute costs at inference. Scaled WS does not have this property: it can be completely “constant-folded” at test time, and is accordingly free at inference (as with BatchNorm).
>
> BatchNorm can incur substantial memory overhead and can be expensive to compute when using large image sizes. This has previously motivated such works as “In-Place Activated BatchNorm”, for tasks such as semantic segmentation where this is an issue. Scaled WS does not have this disadvantage, as it operates on the weights, which for ConvNets tend to be small relative to the size of the activations. We have added new experimental results to our paper (mentioned in our top-level response) comparing training speed to demonstrate the computational advantage of Scaled WS over BN.
>
>
> > “why should a network without normalization mimick the SPP trend of ResNet”
>
> The choice of how signals should propagate in unnormalized networks is largely one of design. In Appendix G.2, we note that we initially explored designing networks to have constant variance, which without prior knowledge one might assume to be a superior choice. We found that such networks were not as performant, and reasoned that mimicking a signal propagation template which we know to work well was a good design choice, as is supported by our experiments.
>
> > “How did the authors compute gamma in eq.(3) in a training phase? Why the provided code contains a learnable gamma?”
>
> Gamma is a nonlinearity-specific scalar constant. It is not learnable, but is a fixed scalar derived similar to the constants from He Initialization, in order to make a nonlinearity approximately variance-preserving. See section 4.3 in the paper for more discussion on this. As mentioned in the text, gamma values can be computed analytically if the form of the variance is known when assuming the distribution of the input, or approximated empirically using the code provided in Appendix D.1. The provided code does contain a zero-initialized scalar gain at the end of a ResNet block, in order to allow one to implement SkipInit. As mentioned in the paper, this is one of several relaxations we employ which we find to work well empirically.
>
> > “Did the authors use trained ResNetV2-600 or randomly-initialized model?”
>
> The SPPs are presented for models at initialization. We have clarified this in the paper.
>
> > “How does SPP goes on with the post-activation network which uses the original bottleneck block consists of Conv-BN-ReLU?”
>
> We have now added the SPP for post-activation ResNets and ResNets with FixUp in the appendix section F (figures 10 and 11).

---

> > ### Author Response · Authors · 2020-11-16
> > **Response to R1, part 2 of 2**
> >
> > > “Why "zero padding" at p.5 affects the variance decay in the rightmost graph in Figure 2?”
> >
> > Zero padding causes signal attenuation by changing the statistics of the input. For example, a given 7x7 tensor drawn from N(0,1) will have a variance of around 1, but a 3x3 convolution operating on this tensor will actually see the 9x9 padded tensor in practice. This 9x9 padded tensor will have 32 elements which are guaranteed to be zero, meaning that the variance of the outputs will be smaller than expected. For activations with large spatial extent, this is negligible, but for the 7x7 activations in the final stage of the network this effect becomes visible in the SPP. For normalized networks, this padding will not have an effect on the signal magnitude because the normalization layers directly enforce unit variance regardless of input variance.
> >
> >
> > > “Please clarify why RegNetY-400MF chose it as the baseline.”
> >
> > We find that NF-EfficientNets do not attain particularly high accuracy because weight standardization interacts poorly with depthwise convolutions. WS restricts the number of degrees of freedom of a filter (an output unit of the layer), which is not a problem for normal full convolutions with many degrees of freedom, but for a depthwise convolution (where each unit has very few degrees of freedom) this substantially reduces capacity. We investigated this empirically by training NF-EfficientNets without WS on the depthwise convs; these models tend to become unstable, but the ones that survived training tended to achieve higher (albeit still not competitive) accuracy.
> >
> > The switch to RegNet models is thus motivated by the fact that these models make use of grouped convolutions, which would not suffer so much from weight standardization. The tuning we do for our models is simply to adopt the EfficientNet scaling laws and other best practices from the literature, which were simply absent from the original RegNet models and training setup, and apply them largely without modification; please see our top-level response for additional motivation behind this choice.
> >
> > Please note that we do not omit comparison to RegNets--they are included in both table 3 (appendix A) and our Accuracy vs FLOPS figure (figure 3) --- but we focus on comparison to EfficientNets as RegNets performance is substantially lower.
> >
> >
> > > “How can the resolution downsampling block in a ResNet affect averaged channel mean and variance?”
> >
> > In a preactivation BN-ResNet, the skip-path takes in a batch-normalized input, so regardless of the variance of the input, the variance of the skip path will be reset. We mimic this property in NF-ResNets by downscaling the skip path by the expected standard deviation of the input.
> >
> > > “Why the ResNet experiments are done with weight decay of 5e-5?”
> >
> > We have re-run these experiments with weight decay set to 1e-4 and have not found a difference in results (with 5e-5 obtaining better results for BN-ResNet-50). We simply chose this weight decay to be consistent with the rest of our experiments. We also note that in our newly added microbatch training experiments, weight decay of 1e-4 further degrades the performance of batch-normalized ResNets when trained with batch sizes of 4 or 8 (by around 3% absolute top-1), so we have retained the setting of 5e-5 for those experiments as well.
> >
> >
> > > “Comparing EfficientNets that are trained without CutMix and Mixup (but used randaug or autoaug) with the proposed models with cutmix and mixup seems to be not fair.”
> >
> > RandAugment is generally a more effective augmentation strategy than CutMix+MixUp. Below we have included numbers for running EfficientNets with CutMix+MixUp, which are worse than the reported numbers for AA/RA (and for the larger variants may even degrade performance relative to the unaugmented baseline without EfficientNet-style aggressive early stopping).
> >
> > EfficientNets: ImageNet top-1 with CutMix+MixUp vs AA/RA (AA/RA numbers from https://github.com/tensorflow/tpu/tree/master/models/official/efficientnet),
> >
> > Variant | CutMix + MixUp | AA or RA
> >
> > B0        |         76.57         |      77.1
> >
> > B1        |         78.94         |      79.1
> >
> > B2        |         79.83         |      80.1
> >
> > B3        |         81.41         |      81.6
> >
> > B4        |         81.84         |      82.9
> >
> > B5        |         82.81         |      83.7

---

### Author Response · Authors · 2020-11-16
**Top-Level Response**

We thank the reviewers for their time, effort, and valuable insights. We have updated the paper draft, incorporating their excellent feedback. Specifically:

1.  Downstream tasks: Two reviewers requested additional experimental results on downstream tasks. We have now run experiments comparing our NF-ResNet backbones against BN-ResNet models for segmentation on Pascal VOC using FCNs [1] following the fine-tuning procedure as used in MoCo [2]. We have also run experiments on depth estimation on the NYUv2 dataset. We have added these results to appendix section H. Our normalizer-free models attain comparable accuracy to batch-normalized models on these downstream tasks.

2. Comparison to prior work: Reviewer 3 requested comparisons to prior work on removing BatchNorm. We have now run experiments with Fixup [3] and SkipInit [4] on ResNets both with and without additional regularization (stochastic depth and dropout), and added these results to Table 1 in the paper. Our proposed NF-ResNets outperform both Fixup and SkipInit.

3. Training speed comparison: We have added training speed numbers (in terms of training iterations/s) to the appendix section A.2 (tables 4 and 5) for BN-ResNets, NF-ResNets, BN-EfficientNets, and NF-Nets, measured on a single V100 across multiple batch sizes. We show that in all cases, NF-ResNets and NF-Nets speed up training over their batch normalized counterparts.

4. Small batch experiments: Two reviewers requested results in the small batch setting, where BN is known to perform poorly[5]. We have now run “microbatch” experiments with batch sizes of 4 and 8 trained for 15 epochs (4.8M and 2.4M total training steps, respectively, the largest we could run in a reasonable number of GPU-days), and added these results to table 2 (section 5.1). The batch-normalized models undergo severe training degradation (especially at batch size 4), while the normalizer-free models retain the same accuracy as when training for the same number of epochs with BS=1024, with no difference in performance for NF-ResNets at BS=4 vs BS=8.

5. On the design of NFNets: Our experiments have two goals. The first is to first demonstrate that we can obtain competitive performance with well-tuned batch-normalized models in a standard ResNet setup. The second goal is to demonstrate that networks without activation normalization can be trained to a performance competitive with the modern state-of-the-art. As weight standardization appears to interact poorly with depthwise convolutions, simply applying our approach to EfficientNets is not sufficient for this purpose, hence why we selected RegNets (which use grouped convolutions) as a baseline model, to which we then simply apply existing best practices (e.g., we use the EfficientNet compound scaling width + depth multipliers, borrowed entirely without modification, and we use ResNet-D downsampling).

Relative to the level of tuning that has gone into EfficientNets (tens of thousands of device-hours of architecture search), the tuning and model design we have done is relatively limited, instead focused on adopting existing modifications which are known to work well. Given the massive amount of tuning which has gone into the top-performing batch-normalized models, it is not unreasonable to assume that attaining competitive performance after removing BN should require some degree of tuning. We feel that it is important to do the necessary work to push our networks towards performance competitive with the state of the art to show the community that high performance is possible without BatchNorm.


[1] Ross Girshick, Jeff Donahue, Trevor Darrell, and Jitendra Malik. Rich feature hierarchies for accurate object detection and semantic segmentation. In Computer Vision and Pattern Recognition, 2014

[2] Kaiming He, Haoqi Fan, Yuxin Wu, Saining Xie, and Ross B. Girshick. Momentum contrast for unsupervised visual representation learning. In Computer Vision and Pattern Recognition, 2014

[3] Hongyi Zhang, Yann N. Dauphin, and Tengyu Ma. Fixup Initialization: Residual Learning Without Normalization. International Conference on Learning Representations. 2018.

[4] Soham De and Sam Smith. Batch normalization biases residual blocks towards the identity function in deep networks. Advances in Neural Information Processing Systems 33 (2020).

[5] Yuxin Wu and Kaiming He. Group Normalization. ECCV 2018.

---

> ### Author Response · Authors · 2020-11-16
> **The Trouble with BatchNorm**
>
> Thanks to the reviewers for pointing out the lack of citations for BatchNorm introducing bugs; this was an oversight on our part. Here we enumerate the documented cases we are aware of, and in the updated draft we have included reference to [3], an academic paper on automated testing which discovers 2 BatchNorm bugs in Keras and one in CNTK.
>
> One example is that a long-standing bug in certain versions of Keras has made it so that even if a user sets the BatchNorm layers into testing mode (as is common when freezing the layers for fine-tuning for downstream tasks) the BatchNorm statistics will continue to update [1][2][3], which is both incorrect and contrary to user expectations. As documented in [3], this is actually the second such bug to be present in a public release of Keras. [3] also documents an incorrect implementation of BatchNorm in the CNTK toolkit.
>
> Another example is that both DCGAN and SAGAN reported results and released code where BatchNorm was run in training mode at test time [4][5], meaning that results improperly depend on the batch size used to generate the samples, something which may stymie an unaware person’s attempt to reproduce said results.
>
> Subtle differences in BatchNorm implementations can also hamper reproducibility. For example, the EfficientNet training code uses a form of cross-replica BatchNorm where the number of devices used to compute statistics varies nonlinearly with the total number of devices [6], meaning that exact reproduction most likely requires access to exactly the same TPU hardware used. Additionally, the EfficientNet code takes a Polyak average of the running BatchNorm statistics, meaning that it takes a moving average of a moving average, further adding to the potentially unexpected interactions.
>
> Breaking independence between training examples specifically causes issues in contrastive learning setups like SimCLR [7] and MoCo [8], which find that BatchNorm requires specific care to prevent information leakage. MoCo accomplishes this by shuffling examples between devices to compute statistics, again adding implementation complexity and hampering reproducibility in the case where one does not have access to the same hardware.
>
>
> [1] https://github.com/keras-team/keras/issues/7051
>
> [2] https://github.com/keras-team/keras/issues/12400
>
> [3] https://www.cs.purdue.edu/homes/lintan/publications/cradle-icse19.pdf
>
> [4] https://github.com/brain-research/self-attention-gan/commit/7702dc5b5f7c58c14c860232505a6e18f8fb720d
>
> [5] https://github.com/Newmu/dcgan_code/blob/master/faces/train_uncond_dcgan.py#L111 and https://github.com/Newmu/dcgan_code/blob/master/lib/ops.py#L52-L57, note that no `u or s values are passed into the batchnorm op, meaning that stats are not accumulated
>
> [6] https://github.com/tensorflow/tpu/blob/master/models/official/efficientnet/utils.py#L124-L148
>
> [7] He, Kaiming, Haoqi Fan, Yuxin Wu, Saining Xie, and Ross Girshick. "Momentum contrast for unsupervised visual representation learning." CVPR 2020
>
> [8] Chen, Ting, Simon Kornblith, Mohammad Norouzi, and Geoffrey Hinton. "A simple framework for contrastive learning of visual representations." arXiv preprint arXiv:2002.05709

---

### Author Response · Authors · 2020-11-23
**Final Update**

We have uploaded a final minor revision of our paper. The first change is that we have decided to change the name of our models from “NF-Nets” to “NF-RegNets,” which we felt was better scholarship and more accurately attributes the lineage of our models. Second, we have added an additional paragraph at the end of the experiments section motivating our choice to tune the design of our NF-RegNet baseline, in line with our comments below.

---

### Comment · ~Lei_Huang1 · 2021-01-19
**Good work for training without activation normalization, with one closely related work missed in the submission**

Good work for training without activation normalization. However, I believe this paper should discuss the differences to the Centered Weight Normalization (CWN) paper[1], which enjoys the nearly same formulation as the proposed Scaled Weight Normalization (SWN) (contrast Eqn.5 in [1] to Eqn.3 of this paper). Noting that $\sigma_w \sqrt{N} = \|| w- \mu_w \||$, and the WS method (Qiao 2019) doesnot divide  $\sqrt{N}$ (Not like CWN and SWN described in Eqn.5). Besides, the idea to set initial scale $\gamma$ is similar, both for maintaining the equal-variance property among layers during training in theory (this paper mainly considers the residual scenario).

[1] Centered Weight Normalization in Accelerating Training of Deep Neural Networks, ICCV 2017

---

> ### Comment · ~Kai_Hu2 · 2021-02-06
> **Another related work**
>
> Hi, we will be very thankful if the authors could consider our work as a related work:
>
> Shao J, Hu K, Wang C, et al. Is normalization indispensable for training deep neural network?. Advances in Neural Information Processing Systems, 2020, 33.

---

### Decision · Program_Chairs · 2021-01-07
**Final Decision**

**Decision:**

Accept (Poster)

**Comment:**

This paper analyses the signal propagation through residual architectures; then suggests a scaling method which, together with weight standardization, allows to train such networks to high accuracy with batch-norm; it demonstrates that the method performs better than previous methods (Fixup, SkipInit), and can be used on more advanced architectures.

The reviewers initially had several concerns, but after the author's revision, these concerns were addressed and most reviewers recommended acceptance. One reviewer did not respond, but I think these concerns were addressed. I think it will help to further convince the readers on the usefulness of the method readers if the authors would check the sensitivity to the learning rate with the current method and compare with other methods (SkipInit, Fixup, BN). The reason I'm suggesting this is that I think one of the main reasons BN is still in popular use is that it commonly tends to make training more robust to changes in hyper-parameters, such as the learning rate (while other methods, like SkipInit and Fixup, require more hyper-parameter tuning).

Overall the analysis and the suggested method seem useful, especially at a small batch size and the writing is mostly clear, so I recommend acceptance.